# Development and psychometric evidence of the Academic Engagement Scale (USAES) in Mexican college students

**Lizeth Guadalupe Parra-Pérez** [1]*, **Angel Alberto Valdés-Cuervo**[1], **Maricela Urías-Murrieta**[1], **Reuben Addo**[2], **Laura Violeta Cota-Valenzuela**[1], **Fernanda Inéz García-Vázquez**[1]

**1** Department of Education, Technological Institute of Sonora, Ciudad Obregón, Sonora, México,
**2** Department of Social Work, Fresno State University,Fresno, California,United States of America

\* lizeth.parra13804@potros.itson.edu.mx

## Abstract

School engagement is considered an effective college dropout antidote; therefore, understanding the construct, its underpinnings, and its effects remains critical for scholars. Although several scholars have offered multiple scales to measure engagement, their use has been hindered by significant limitations. This study sought to develop a scale to measure academic engagement by unifying and improving existing work and theories that resulted in a three-dimensional measurement model (behavioral, emotional, and cognitive). The items included were validated by a group of experts who ensured that the wording of the items captured the uniqueness of the college experience. A sample of 992 Mexican college students was used to test the fit of a second-order three-dimensional factor model of school engagement. The sample was randomly split in two for model cross-validation. Confirmatory factor analyses confirmed that student engagement is a three-dimensional construct, with evidence that supports the hypothesized second-order engagement factor structure (behavioral, emotional, and cognitive). The stability of these models was confirmed by using an independent sample. Measurement invariance by gender was found in this model. Then, differences in latent factor means were analyzed. Finally, the scale showed discriminant and concurrent validity. These results indicate that the scale is theoretically and psychometrically grounded for measuring college students' school engagement.

## Introduction

Endemic issues in higher education, such as low achievement and high dropout rates, have proven difficult to eradicate using simple school-based interventions. The academic engagement has risen as a protective factor against these issues [1–3]. Although there is no consensus on the definition of student engagement, scholars agree that it refers to the extent of students' attitudes and involvement in school-related learning activities [4, 5]. Although the positive effects of academic engagement on performance and attainment have been widely studied [6–8], what perhaps should call the most attention is its potential as an antidote against dropouts, which can be prevented through interventions by the school community [9–11]. Hence,

**Funding:** This research was financed by the Technological Institute of Sonora through the budget for Competitiveness and Academic Capacity Project 2023-0303.

**Competing interests:** The authors have declared that no competing interests exist.

describing and understanding student academic engagement are two crucial goals for researchers and policymakers. However, efforts to do so have been undermined by considerable variations in the conceptualization and measurement of engagement. This condition makes it hard for scholars to choose the most appropriate instrument and evolve the current understanding of the construct [12, 13].

An important long-lasting issue is that multiple terms are used interchangeably to name the construct (e.g., student engagement, school engagement, academic engagement, class engagement, and schoolwork engagement). Such inconsistencies lead to multiple conceptualizations of engagement, reflected in various measurement models, avenues for collecting data, assessment focus (school or learning), and variation in the content of items used in each instrument. Moreover, most current scales tend to capture factors that affect school success (e.g., school goals and self-regulation) rather than indicators of engagement per se. Additionally, these scales often need to define the object in which students engage (e.g., school activities, homework, or learning activities). In line with the above, Finn and Zimmer [6] emphasized that most scales include items that blur the lines between engagement, antecedents, and consequences of engagement; that is, they lie outside the limits of the concept.

### Theoretical approach for measurement academic engagement

Current measurement variations in school engagement are due to falling under various perspectives and serving different purposes [12, 14–16]. In the current literature, two overarching theoretical approaches guide most studies in higher education: behavioral and psychological. The following sections analyze students' self-reported measures of engagement development based on these perspectives.

### Behavioral approach

The National Survey of Student Engagement (NSSE), formerly known as the College Student Engagement Questionnaire (CSEQ), is by far the most popular instrument for measuring academic engagement in higher education students based on the behavioral perspective [17–19]. Although several studies confirm the usefulness of the NSSE in measuring the construct, some scholars [4, 20–22] argue that the questionnaire is open to criticism, alluding to at least three primary problems. First, the NSSE has an overly broad definition of student engagement and lacks a specific theory that justifies items. Then, a broad range of student performance and outcomes should be included in the scale (e.g., working for a pay-off campus, relaxing, and socializing with friends). Second, several scholars [23–27] have explored the test structure using EFA and CFA, finding that NSSE's structure did not hold in the U.S. and Canadian samples. For example, using the NSEE, scholars have found eight factors [23], whereas others have reported a three-factor internal structure [28]. Although it is relatively common that the scale's internal structure varies across different samples, its invariance helps compare the results of the studies on the subject. Third, the NSSE needs to be narrower in scope because it assesses many educational experiences, not precisely engagement. Fourth, we do not know of studies that examined the NSSE measurement invariance by sociodemographic variables, such as gender and race.

### Psychological approach

The Utrecht Work Engagement Scale [29, 30] was designed for work settings and adapted for educational purposes through three dimensions: absorption, dedication, and vigor [31–34]. This model is aligned with psychological perspectives and portrays engagement as an internal process while individuals carry out job-related activities. Under this model, studying positive-

oriented human and psychological resources remains critical for improving work performance. Although current evidence suggests that the UWES-S presents acceptable psychometric properties, even in Latin American populations [31, 35], scholars warn about two potential issues for its usage in the educational context. First, the UWES-S does not collect critical information about school attendance, such as adherence to classroom norms, following rules, and respect for teaching staff [36]. Second, the items were developed to measure engagement within the work context; therefore, there are concerns about the adequacy of rephrasing items to measure this construct within the university context [3, 37].

The affective, behavioral, and cognitive (ABC) model of attitudes [17] confirmed that the consistency of responses toward the three components might be considered an index of the measured attitude and suggest that each component has distinct antecedents and consequences from those of the others. In line with this framework, other scholars [4] proposed a renewed engagement model that focuses on the emotional processes and cognitive and behavioral efforts invested throughout the learning process and participation in school activities.

Although inconsistencies in the definitions and number of dimensions of engagement under this model persist, an essential commonality throughout the literature is that engagement is a multidimensional construct [6, 13, 38]. However, several studies [1, 39–41] report mixed results about its internal structure. As a result, engagement is often measured as a two, three, or even four-dimensional construct. For some scholars [14], engagement is a two-dimensional model of emotional and behavioral aspects. As for others [4], engagement is a three-dimensional model comprising cognitive, emotional, and behavioral dimensions. On the other hand, Wang et al. [42] assert that engagement is a four-dimensional model that includes cognitive, behavioral, emotional, and social engagement.

The work conducted by these scholars attempts to fill the current gaps in available scales; however, more efforts are needed. Another essential concern is the alignment of the engagement scope and the unique context in which the population is tested [12–15, 42–45], given that engagement is both situational and responsive to the environment [4, 44]. It arises from the interaction of context and individual; scholars must ensure that the scales capture student engagement within their specific contexts. In other words, scales comprising items may need to be more appropriate, especially if they are interested in exploring how much engagement varies under specific contextual factors. Thus, scale items must be carefully worded to measure engagement in the specific context of higher education [12, 14, 15, 42–45].

Based on research conducted by other scholars [6, 12–14, 43, 44], we believe that the current scales of student engagement present at least four critical issues. First, they fall under various perspectives and serve different purposes; therefore, items from the current scales might only partially be accurate for testing student engagement in Mexican universities. Second, in addition to lacking theoretical guidance, some items in the current scales capture factors that affect engagement (antecedents and consequences) rather than indicators of engagement per se [6, 13, 43]. Third, given that engagement is both situational and responsive to the environment [4, 44], there is an urgency in aligning the scope of student engagement with the unique context in which the population is tested. Moreover, disagreement over the dimensionality of the engagement measure persisted.

Considering the above, as our colleagues [43], we believe appropriate instruments to measure student engagement at different levels and contexts are needed, given that even theory-based and well-thought-out scales have complicated this distinction [13, 43]. Therefore, developing theoretically grounded, culturally sensitive, and robust psychometric scales is necessary to advance current research and inform practitioners and policymakers accurately.

## Gender differences in engagement

Another essential issue that the literature still needs to address is the need to prove that engagement measurements function similarly across males and females. Although prior research indicates that levels of student engagement may vary significantly by gender [46–48], some research continues to report mixed results. Whereas a group of scholars [49] found that females are equally engaged as their male counterparts, a recent study [50] reported that males participate more in classes than females. Although several factors may explain such mixed results, it is essential to underline that these findings came from studies that did not prove measurement invariance in the scale used. Thus, such analyses make gender comparisons of engagement inaccurate or dubious at best [51]. Therefore, ensuring scale invariance between both groups is necessary to increase confidence that the levels of engagement can be compared in future research [52].

## Engagement relationships with external variables

Academic rigor demands that students invest time in their schoolwork and actively learn meaningful content with high-order thinking at an appropriate level of expectation in a given context [53–56]. Academic rigor is greater when learners are taken from mere comprehension toward higher-order thinking through academic activities and dynamics. Academic rigor is a variable influencing academic engagement; it is reflected in higher-order activities that encourage student psychological engagement because it encourages student self-regulation over their learning actions [57, 58]. Therefore, rigorous academic environments are expected to encourage students to take ownership of their learning and build commitment to in-depth learning. Overall, current literature suggests that student engagement can be influenced by rigor embedded in academic-related activities. Some studies [53, 54, 58] have provided evidence that academic rigor significantly predicts student engagement. Likewise, other studies [56, 58] have reported that academic rigor is a mediating factor in student engagement, which deserves further exploration.

## The present study

We adopted a psychological approach, given that our study is interested in the psychological processes in students' minds and what they think and feel. We foresee our work as an effort that goes beyond "what is observable" and what is measurable. This clarity in our study allowed us to consider the psychological over the behavioral perspective. Based on this framework, this study aims to develop and examine the psychometric properties of the University Students' Academic Engagement Scale (USAES) from a three-dimensional psychological perspective [4]. While the literature review provided the theoretical basis for defining the domain and its dimensions, qualitative techniques allowed us to move from the abstract point to its manifest form. Thus, to ensure that the University Students' Academic Engagement Scale (USAES) items effectively capture the lived experiences of Mexican college students, we conducted focus group work. During the item-generation process, we combined inductive and deductive methods [59–62]. In order to have a diverse sample of students across the country, we met students and faculty members from three public universities located in the north (Sonora), center (Nayarit), and south (Chiapas) of the country (three groups of students and three groups of teachers from each university) who volunteered to participate. Our meetings with students included 36 (20 female and 16 male) research participants (12 from each university) from the arts and humanities, engineering, and natural sciences. Similarly, our meetings with faculty members included 18 (11 female and 7 male) research participants (six from each state) who teach social science, engineering, and natural sciences.

Before the focus group discussions, we informed the research participants about the meaning of engagement and its three factors (behavioral, emotional, and cognitive). Research participants discussed four themes: (a) how truly engaged students behave in daily school days; (b) how truly engaged students connect with the academic community, their peers, and teachers, and the institution itself when they feel studying college is an excellent opportunity they should take advantage of; (c) how truly engaged students manage to learn as much as possible when they know such learning will be helpful in real life, and (d) whether a person can get behavioral, emotional, and cognitively engaged? We performed a thematic analysis of the focus group conversations. As a result, 25 indicators of academic engagement among Mexican college students emerged to assess the construct in three dimensions (behavioral, emotional, and cognitive).

Subsequently, a group of eight Mexican researchers in higher education and engagement was consulted to ensure that the USAES items aligned with the context and reality of Mexican higher education classrooms. The experts were asked to evaluate the relevance of each item for measuring the behavioral, cognitive, and affective dimensions of engagement on a 4-point rating scale (1 = not relevant to 4 = very relevant). Using the content validity index (I-CVI) method, we experienced a considerable reduction in the item pool. 11 of the 25 original items of the USAES were removed; only 14 items accomplished a content validity index (CVI) greater than .80 [63, 64]. This process improved theoretical support and the appropriate selection of situations to measure student engagement within the Mexican context, respecting its antecedents and consequences.

To examine evidence of the validity of the USAES response interpretation, we addressed the following purposes: (a) assessing the fit of a second-order factor model of academic engagement (see Fig 1); (b) assessing the measurement model replicability in an independent sample (cross-validation); (c) examining the discriminant validity of USAES factors; (d) analyzing the measurement model invariance by sex; (e) comparing latent mean differences across sex, once appropriate measurement invariance is present; and (f) examining the USAES concurrent validity by assessing its associations with a measure of academic rigor.

We hypothesized that: Hypothesis 1 (internal structure): the items used to measure academic engagement display a second-order factor structure that contains three first-order factors: (a) behavioral dimension that includes indicators of positive conduct such as following the rules, adhering to classroom norms, and the absence of disruptive behaviors; (b) cognitive dimension, which includes items reflecting the internal investment of cognitive energy that students exert to attain above the minimal understanding of the course content; and (c) affective dimension, which comprises indicators of involvement by feeling included in the school community as well as feeling the school is a significant part of their own lives. Hypothesis 2 (cross-validation): An independent sample replicates the internal model structure. Hypothesis 3 (discriminant validity): each dimension of student engagement discriminates from the other scale dimensions. Hypothesis 4 (measurement invariance): the scale is an invariant measure across gender. Hypothesis 5 (latent means comparisons): Female students have higher academic engagement levels than male students. Hypothesis 6 (concurrent validity): the dimensions of student engagement (cognitive, affective, and behavioral) are positively associated with academic rigor.

## Materials and methods

### Participants

We used a non-probabilistic approach to collect data from Mexican college students from universities in the States of Nuevo León, Sonora, Jalisco, México City, Guanajuato, and Tabasco.

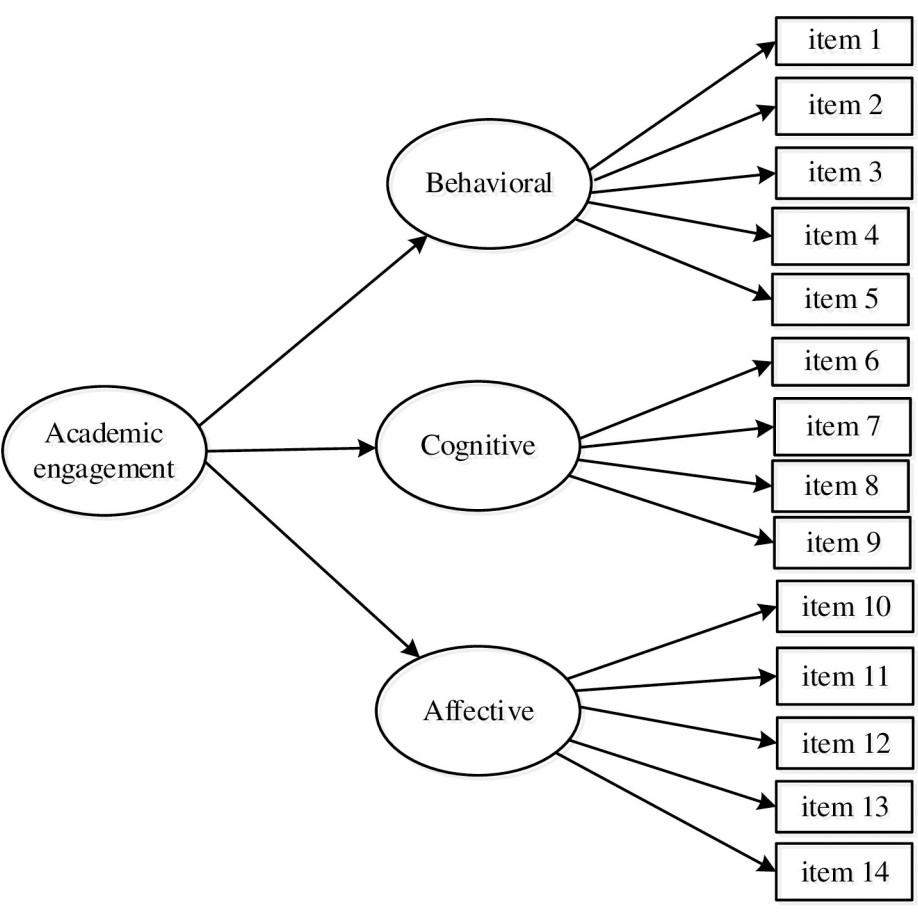

**Fig 1. Factor model of academic engagement depicting a second-order factor model.**

These universities attended to students of all SES, particularly those with low SES and middles-SES. We used quota sampling by sex, ensuring that the proportions reported for 47% of men and 53% of women in Mexican universities [65] were similar to our sample. Additionally, we ensured that our sample included similar proportions of students from different areas of knowledge.

Sample students came from seven public universities across Mexico (Monterrey, Hermosillo, Obregon City, Guadalajara, Mexico City, Guanajuato, and Tabasco). The non-probabilistic sample included 992 students (about 140 from each state), 469 (47%) males, and 523 (53%) females, with ages ranging between 17 and 64 years old ($M$ = 20.6 years, $SD$ = 2.9). The sample included 227 students from natural sciences (22.8%), 226 from biology and health sciences (22.7%), 211 from engineering and technology (21.2%), and 328 from the arts and humanities (33.3%). Students came from different semesters as follows:1st = 145, 2nd = 182, 3rd = 170, 4th = 125, 5th = 248, and 6th = 247. The sample was randomly divided into two subsamples for model calibration ($n$ = 496) and cross-validation ($n$ = 496). All measures were responded to in Spanish.

### Measures

**Academic engagement.** This scale was developed for the study. The USAES comprises 14 items that used a five-point agreement scale, ranging from 0 (*never*) to 5 (*always*), to assess: (a)

*Behavioral engagement*, which comprises participation in academic and non-academic activities and positive conduct (5 items, e.g., "I do attend all my classes, labs, practices, and seminars on time"); (b) *Emotional engagement*, encompasses enjoyment of learning and literary activities (5 items, e.g., "I feel lucky and honored to attend college"); and (c) *Cognitive engagement*, implies the self-regulated aspects of learning, uses of deep learning strategies, and higher order thinking skills (4 items, e.g., "When class topics result in difficulty, I do not give up on them until I grasp them").

**Academic rigor.** The Academic Rigor Scale [66] was used. This scale comprises nine items (e.g., "I have to look for different information resources to accomplish the academic task that my teacher assigns") in a five-point frequency response format, ranging from 0 (*never*) to 4 (*always*). Confirmatory factor analysis (CFA) showed a good fit of the unidimensional model to the data ($SBX^2$ = 40.98, $df$ = 20, $p$ = .004; SRMR = .02; TLI = .97; CFI = .98; RMSEA = .043, 90% CI [.032, .055]. The McDonald's Omega $\omega$ is .88, and the average variance extracted (AVE) is .58. Standardized factor loadings ranged between .62 ($p$ < .001) and .73 ($p$ < .001).

## Procedure

The study was approved by the Ethical Committee of the Technologic Institute of Sonora (Authorization number: PROFAFI 2022 0003). Permission to conduct the study was then granted by university authorities across different universities. We posted and distributed hand-outs seeking volunteers to participate in the study. After obtaining signed consent letters from volunteer participants, they were informed about the research purpose at the beginning of the data collection meeting. The students were informed of the nature of their participation and that they could withdraw from the study at any time. Nine hundred ninety-two students completed the online surveys. All measures were responded to in Spanish.

## Data analysis

The study included all data. The initial sample ($n$ = 992) was randomly split into two subsamples ($n$ = 496), with one sample used for calibration purposes and the second to test the measurement model's replicability. Item score distributions (mean, standard deviation, skewness, and kurtosis) were explored using the *SPPS 26* software. The item values of skew and kurtosis suggest univariate normality [67].

## Dimensionality

Confirmatory factorial analyses were calculated using diagonally weighted least squares (DWLS) estimator available in *Mplus 8* software. Some studies have reported that the DWLS estimator in ordered category variables results in an acceptable parameter for estimates and standard errors [68–70]. To assess the scale dimensionality, we tested the goodness-of-fit of the first-order factor model. After confirming the model's adjustment, we tested a model with three factors as indicators of a second-order factor model (see Fig 1). To evaluate the model's global goodness of fit, we used the $SBX^2$ statistic [68]. The structural equation modeling literature suggests that $SBX^2$ with $p$ associate > .05 indicates that the model fits the data. However, as suggested by some researchers [71, 72], the $SBX^2$ statistic is sensitive to large sample sizes. Therefore, we reported additional fit indices such as the comparative fit index (CFI), standardized root mean square error of approximation (RMSEA), Tucker-Lewis index (TLI), and standardized root mean square error of approximation (SRMR). Based on the structural equation modeling literature [72–74], CFI ≥ .95, and TLI ≥ .90, indicate an excellent model adjustment. For the SRMR and RMSEA, a value < .05 indicates an excellent fit, and values under .08 confirm an acceptable fit.

In order to compare the adjustment of the three first-order factor model and a second-order factor model, we assumed that when differences in $SBX^2$ ($\triangle SBX^2$) are significant, a model with less $SBX^2$ had a better fit to the data [72]. Additionally, we considered differences in the Bayesian Information Criterion ($\triangle BIC$) > 10, suggesting differences in the model fit; a model with less BIC has a better fit [75, 76].

### Discriminant validity evidence

Discriminant validity confirms that the constructs are empirically unique [77]. It also ensures that a latent variable is "not correlated too highly with measures from which it is supposed to differ" [77, 78]. In line with what was suggested in the literature, we assumed that discriminant validity is confirmed when the AVE in each factor is greater than the square of this correlation with the other scale's factors [78, 79].

### Reliability evidence

Evidence of reliability was tested using average variance extracted (AVE) and total Omega coefficient ($\omega$). Values of AVE = .50 and $\omega$. = .70 were considered indicators of adequate reliability [80, 81].

### Model cross-validation

Model replicability was tested by comparing the model adjustment in an independent sample (cross-validation) using the differences in $SBX^2$, CFI, and RMSEA. Based on the literature, we considered that $\triangle SBX^2$ with $p$ associate < .05, $\triangle CFI$ < .001, and $\triangle RMSEA$ < .015 are indicative of model replicability in an independent sample. In the case that statistics differ, we trusted the differences in CFI and RMSEA due to the large sample used in the study [72, 73].

### Measurement invariance by gender

We tested configurational, metric, and scalar invariance across groups. To compare these nested models, we used the difference in $SBX^2$ albeit differences in $SBX^2$ less than the critical value ($SBX^2$ with $p$ associate < .05), which suggests that constraints imposed are equivalent across groups. The $SBX^2$ statistic is sensitive to a large sample. Thus, we used goodness-of-fit indexes, such as $\triangle$ CFI and $\triangle RMSEA$ [82, 83]. This study uses the cut-off rules proposed by these authors, who indicate a $\triangle$ CFI lower than .01, and $\triangle RMSEA$ lower than .015, suggesting model equivalence.

### Latent means comparison

Once the invariance of scale factor loadings and item intercepts were confirmed, the mean differences by gender in behavioral, cognitive, and affective dimensions of academic engagement were tested. A $z$-statistic was used to calculate the differences between latent means. The factor means for females were set to zero, while the groups' factor means for males were freely estimated.

### Concurrent validity

We examined the relationship between academic engagement and academic rigor. Concurrent validity indicates the amount of agreement between two different assessments taken simultaneously [73]. Based on the literature, we expected academic rigor to be positively related to academic engagement. In order to interpret such a correlation, we adopted the guides offered

**Table 1. Means, standard deviations, skewness, and kurtosis of the calibration sample (n = 492).**

| Item | M | SD | Skew | Kurtosis | CVI |
|---|---|---|---|---|---|
| Item 1 | 3.15 | 0.07 | -0.67 | 0.66 | .79 |
| Item 2 | 3.72 | 0.54 | -2.05 | 1.88 | .83 |
| Item 3 | 3.69 | 0.51 | -1.33 | 0.76 | .90 |
| Item 4 | 3.65 | 0.56 | -1.51 | 2.71 | .78 |
| Item 5 | 3.77 | 0.51 | -1.73 | 1.16 | .90 |
| Item 6 | 3.52 | 0.67 | -1.31 | 1.42 | .88 |
| Item 7 | 2,88 | 1.11 | -0.74 | -0.41 | .85 |
| Item 8 | 2,91 | 1.12 | -0.84 | -0.31 | .80 |
| Item 9 | 2,99 | 1.01 | -0.93 | -0.44 | .90 |
| Item 10 | 3,24 | 0.91 | -1.07 | 0.66 | .83 |
| Item 11 | 2,73 | 0.97 | -0.33 | -0.51 | .87 |
| Item 12 | 2,52 | 1.04 | -0.25 | -0.49 | .79 |
| Item 13 | 3,13 | 0.89 | -0.76 | -0.06 | .95 |
| Item 14 | 2,84 | 1.05 | -0.57 | -0.41 | .93 |

by Cohen [84], which suggest that an *r* of .20 indicates a small effect size, *r* = .50 a medium effect size, and a large effect size is indicated by *r* = .80.

## Results

### Analysis of item's distribution

The data in Table 1 show that the students display acceptable academic engagement behaviors. The responses of eight items are centered in the 'frequently' category, while the remaining items are centered in the 'sometimes' category. Skewness and kurtosis values indicated a normal univariate distribution for all items. Content validity index (CVI) values suggest that items are pertinent for measuring the construct.

### Dimensionality analysis

The scale dimensionality was examined in several steps. First, we assessed a three-first-order factor model (behavioral, affective, and cognitive engagement). Second, we modeled the three first-order factors as indicators of a second-order engagement factor. Table 2 shows that the three first-order models fit the data satisfactorily. The positively moderated correlation between first-order factors (.42 to .53) suggests the second-order factor is plausible [72, 73]. CFA confirmed that the second-order factor model fit the data. The goodness of fit of the second-order factor model (Model B) was not statistically better ($\triangle$SBX$^2$) = 8.54, $\triangle df$ = 1, $p$ = .003; $\triangle$BIC = 6.83) than the three first-order factor models (Model A). However, based on this theory, we chose model B for the remaining analyses.

The values of standardized loadings for the one second-order ranged from .72 to .88 ($p <$ .001). Also, all standardized factor loadings for first-order indicators of observable indicators

**Table 2. Goodness-of-fit statistics of the hypothesized three first-order factors and one second-order factor model (n = 492).**

| Model | SBX$^2$ | df | p | SRMR | CFI | TLI | RMSEA | BIC | $\triangle$SBX$^2$ | $\triangle df$ | $\triangle$BIC |
|---|---|---|---|---|---|---|---|---|---|---|---|
| A. Thee first-order factors | 110.07 | 72 | .003 | .04 | .95 | .94 | .04 [.02, .06] | 355.15 | | | |
| B. One second-order factor | 101.53 | 73 | .015 | .03 | .95 | .93 | 05 [.03, .07] | 348.32 | 8.54 | 1 | 6.83 |

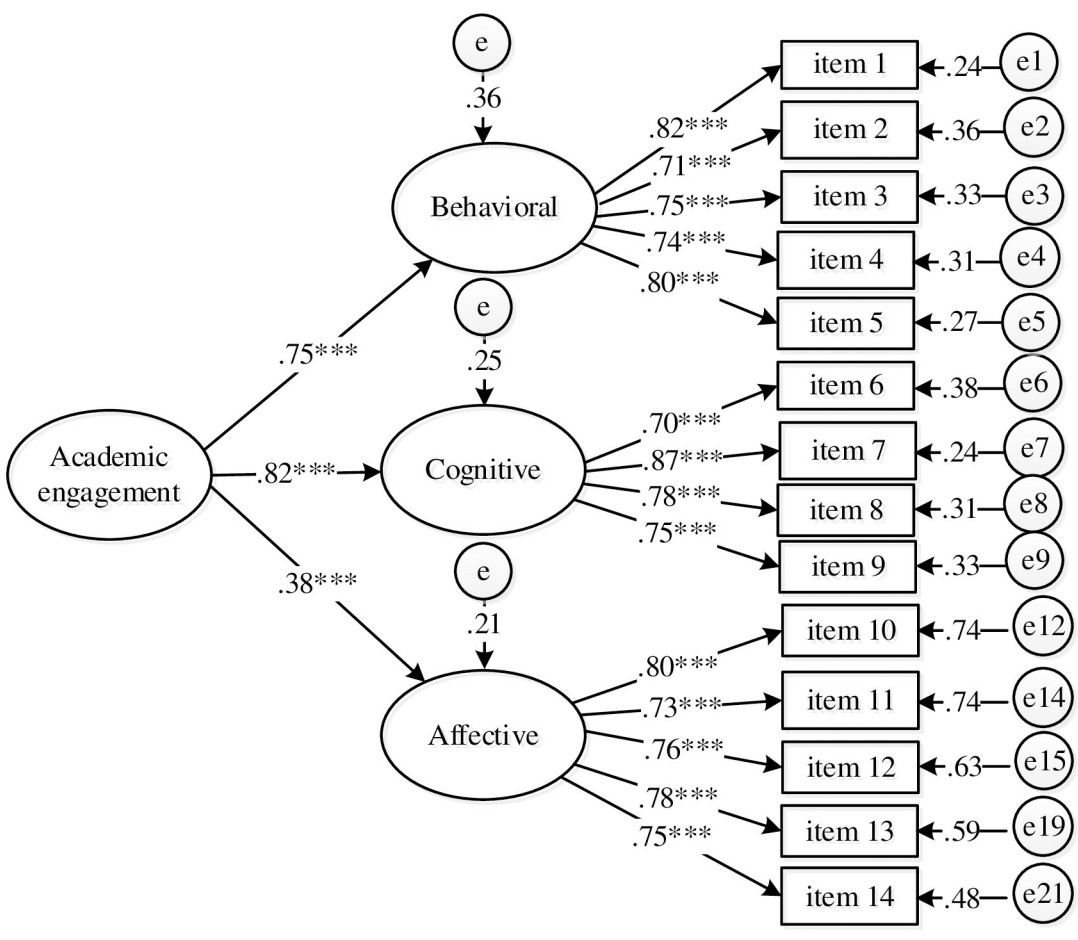

**Fig 2. Second order model.**

ranged between .61 and .84; thus, they are statistically significant ($p <$.001) (see Fig 2). The reliability of the second-order factor (AVE = .65, $\omega$ = .84) and first-order latent models was adequate: behavioral engagement (AVE = .52, $\omega$ = .84), affective engagement (AVE = .54, $\omega$ = .85), and cognitive engagement (AVE = .54, $\omega$ = .82).

## Cross-validation

A multi-group procedure was used to test the replicability of the measurement model in an independent sample of university students ($n$ = 492). The results confirm the configural (SBX$^2$ = 203.56, $df$ = 146, $p$ = .001; SMRM = .032; CFI = .95; TLI = .92; RMSEA = .039, 90% CI [.034, .044]), metric, and scalar invariance in both samples (see Table 3). These analyses support the replicability of the model.

**Table 3. Goodness-of-statistic for testing model invariance across calibration sample (n = 492) and validation sample (n = 492).**

| Model | SBX$^2$ | df | $\triangle$SBX$^2$ | $\triangle df$ | p | $\triangle$CFI | $\triangle$RMSEA |
|---|---|---|---|---|---|---|---|
| Configural | 203.56 | 146 | | | | | |
| Metric | 225.46 | 157 | 21.90 | 11 | .025 | .003 | .001 |
| Scalar | 236.15 | 160 | 32.59 | 14 | .003 | .005 | .002 |

**Table 4. Summary of fit statistics for testing measurement invariance by gender of one second-order dimensional model of academic engagement (N = 984).**

| Model | SBX$^2$ | df | △SBX$^2$ | △df | p | △CFI | △RMSEA |
|---|---|---|---|---|---|---|---|
| Configural | 198.99 | 146 | | | | | |
| Metric | 222.93 | 157 | 23.94 | 11 | .013 | .004 | .001 |
| Scalar | 230.75 | 160 | 31.76 | 1 | .004 | .006 | .001 |

**Table 5. Summary of fit statistics for testing measurement invariance by gender of one second-order dimensional model of academic engagement (N = 984).**

| Factor | $M_{diff}$ | z-statistic | p | Cohen's d |
|---|---|---|---|---|
| Behavioral | -0.52 | -6.21 | <.001 | 0.41 |
| Affective | -0.13 | -1.66 | .097 | 0.11 |
| Cognitive | -0.11 | -1.08 | .914 | 0.001 |
| General engagement | -0.29 | -.3.11 | <.001 | 0.20 |

## Measurement invariance

The configural model fit to the data (SBX$^2$ = 198.99, df = 146, p = .002; SRMR = .05; CFI = .96; TLI = .95; RMSEA = .05, 90% CI [.024, .064]). The fit indices confirmed metric and scalar invariance (see Table 4). Overall, the results suggest that the scale is psychometrically consistent in measuring academic engagement in both genders. Therefore, this construct can meaningfully assess and compare the groups.

Due to the confirmed scalar invariance, we compared the latent means between males and females. The results showed greater behavioral and general engagement in females than in males. Additionally, no other differences in affective or cognitive engagement between genders were confirmed (see Table 5).

## Discriminant validity

The discriminant validity was tested to examine the uniqueness of the engagement factor. For all factors, the AVE is greater than the square of the correlation with another factor, confirming each factor's uniqueness (see Table 6).

## Concurrent validity

We found significant positive correlations between the behavioral, affective, and cognitive dimensions of academic engagement and academic rigor (see Table 6). The effect size of these correlations suggests the practical and theoretical implications of the results. Overall, these results support the scale's concurrent validity.

**Table 6. Correlations between academic engagement dimensions and academic rigor.**

| Factor | Behavioral | Affective | Cognitive | General Engagement | Academic rigor |
|---|---|---|---|---|---|
| | AVE = 52 | AVE = 54 | AVE = .54 | AVE = .65 | AVE = .58 |
| Behavioral | - | | | | |
| Affective | .42*** (.18) | - | | | |
| Cognitive | .35*** (.12) | .54*** (.29) | - | | |
| General engagement | .54*** (.29) | .69*** (.47) | .66*** (.43) | - | |
| Academic rigor | .43*** (.18) | .64*** (.41) | .62*** (.38) | .53*** (.28) | - |

***p < .001

## Discussion

Students' engagement in academic settings diminishes long-lasting educational issues such as low achievement and high dropout rates. Consequently, many scholars have been interested in exploring this construct. Although several scales have been developed to explore the construct, its underpinning, and effects present important limitations that prevent its use in future research. Therefore, we sought to develop and validate a scale that accurately captures the engagement of Mexican students by unifying and improving existing work and theories in the field. The USAES items were validated by a group of experts who ensured that the wording captured the uniqueness of the college experience. Together, the improvements in this new scale have a wide range of possible applications for the instrument, which may contribute to advancing research in the field. Overall, the results indicate that the second-order measurement model fit the data. Furthermore, the results supported measurement invariance, indicating that the scale is equivalent by gender. Finally, the results confirm the discriminant and concurrent validity of the USAES.

### Student engagement as a second-order factor

The results confirm that the USAES presents, as hypothesized, a second-order structure comprising three first-order factors (behavioral, emotional, and cognitive). The 14 items comprising the scale are grouped into three first-order factors according to psychological perspective [4]. These findings align with previous research [2–4, 12], indicating that student engagement has a second-order structure, with responses to the measurement of engagement grouped through three first-order factors (behavioral, emotional, and cognitive). Moreover, discriminant validity proved that each engagement factor (behavioral, emotional, and cognitive) assesses a different manifestation of engagement; these results align with previous studies [2, 12, 48]. Our findings suggest that focus group work led to the development of items for each engagement factor that effectively capture the attitudes and behaviors that reflect school engagement in college students.

### Measurement invariance by gender

These results provide empirical evidence that supports the measurement invariance of student engagement by gender. Our findings indicate that the scale is equivalent to the construct for both genders. Therefore, unlike other scales, the USAES allows scholars to compare genders with more validity. Given that both factor loadings and interceptors resulted in invariance, we examined the latent mean differences in second-order factors (behavioral, emotional, and cognitive engagement). As other scholars [2, 12, 48] reported, our results from mean scores indicate that females have higher levels of behavioral engagement than males. Consistent with previous research [49], we found no significant differences between males and females regarding emotional and cognitive engagement. Regardless of these results, future research should address gender differences using valid-invariant scales. Indeed, such research would shed light on the blurred relationship between gender and school engagement. Research may contribute to a better understanding of the underpinnings of school engagement and its effects on students.

### Evidence of external validity

Our findings confirm what other scholars [1, 4, 6, 56, 57] have consistently suggested: the academic culture significantly influences students' engagement. In addition, the data provide evidence supporting concurrent validity. Specifically, our results suggest a positive association between academic rigor and engagement, a relationship theoretically proposed in the literature

yet poorly explored [2]. Therefore, this finding is relevant because it supports theorists who assert that academic rigor, reflected in higher-order thinking activities, can encourage student engagement.

## Theoretical and practical implications

These results have substantive methodological implications for scholars interested in school engagement. Similar to other studies [4, 12, 13], our results confirm that school engagement is a second-order factor comprising three first-order factors: behavioral, emotional, and cognitive. Even more critically, they confirm the value of the three-dimensional model proposed by Fredericks et al. [4] and its replicability across different populations. The study also confirmed the value of using well-focused and contextualized items to assess school engagement. However, future research should be conducted using invariant scales and focus on examining the factors that lead individuals to different types of engagement (behavioral, emotional, and cognitive) and their effects on students in and out of school.

From a practical perspective, using the USAES, scholars may identify the underpinnings of school engagement, which can be targeted with future school interventions to benefit students. School interventions are supported using the USAES because they provide more accurate and valid inferences. Likewise, it is an instrument that may be used to identify which type of engagement (behavioral, emotional, and cognitive) brings better effects on students, in and out of school, to intervene accordingly. More importantly, the USAES may provide valuable information to administrators and policymakers to implement more effective educational programs.

## Limitations

Some limitations of this study should be noted. First, the study design (cross-sectional) needs to assess students' engagement throughout the college experience. Thus, longitudinal studies are necessary to examine changes in engagement over time. Second, the results rely on self-reported data; personal biases may have influenced students' responses. Further studies should explore the relationships between USAES responses with measures that include different data sources (e.g., teachers or peers) and measurement methods (e.g., observation) to expand the scale's usefulness in measuring academic engagement among university students. Moreover, even though we made a considerable effort to replicate the study population's structure to obtain a sample that can be considered representative, we acknowledge a high risk of bias. Third, these findings are specific to different regions of Mexico. It is imperative to acknowledge that students from universities that attend rural and indigenous schools may have different experiences with school engagement. As a result, the conclusions can only be generalized to Mexico's data collection area. Fourth, although we demonstrated measurement invariance by gender, examining the measurement invariance of other sociodemographic variables is necessary to compare other groups. Five, the reported effect size of the relationships of academic engagement´s dimensions and academic rigor is based on Cohen´s guides, but it is arbitrary [85]. These results should be compared with future studies that use similar designs in Mexican adolescent to test more accurately the effect size. Finally, although the USAES was explicitly developed to consider the Mexican context, we suggest cross-cultural studies to assess the replicability of the instrument in other populations.

## Conclusion

This study provides a theoretical and highly scrutinized instrument with robust validity for assessing school engagement as a second-order factor comprising three first-order factors:

behavioral, emotional, and cognitive. The USAES shows improvement in assessing each dimension of engagement, which contributes to identifying the antecedents and consequences of school engagement in students. The USAES is a suitable tool in higher education that should be adapted for use with different Mexican samples of university students and countries. However, such a process requires an examination of scale measurement invariance in these samples.

## Acknowledgments

We would like to express our deep gratitude to the Department of Education of the Technologic Institute of Sonora and the Department of Social Work of the University of California Fresno for their significant support throughout the research process.

## Author Contributions

**Conceptualization:** Lizeth Guadalupe Parra-Pérez, Angel Alberto Valdés-Cuervo, Maricela Urías-Murrieta, Reuben Addo.

**Data curation:** Lizeth Guadalupe Parra-Pérez, Laura Violeta Cota-Valenzuela.

**Formal analysis:** Lizeth Guadalupe Parra-Pérez, Angel Alberto Valdés-Cuervo.

**Funding acquisition:** Angel Alberto Valdés-Cuervo, Maricela Urías-Murrieta, Fernanda Inéz García-Vázquez.

**Investigation:** Lizeth Guadalupe Parra-Pérez, Maricela Urías-Murrieta, Reuben Addo, Laura Violeta Cota-Valenzuela.

**Methodology:** Lizeth Guadalupe Parra-Pérez, Angel Alberto Valdés-Cuervo, Maricela Urías-Murrieta, Reuben Addo.

**Project administration:** Maricela Urías-Murrieta.

**Resources:** Maricela Urías-Murrieta.

**Validation:** Laura Violeta Cota-Valenzuela.

**Writing – original draft:** Lizeth Guadalupe Parra-Pérez, Fernanda Inéz García-Vázquez.

**Writing – review & editing:** Lizeth Guadalupe Parra-Pérez, Angel Alberto Valdés-Cuervo, Reuben Addo, Laura Violeta Cota-Valenzuela, Fernanda Inéz García-Vázquez.

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
