## [Decision Letter · Decision Letter 0]

16 Jan 2023

PONE-D-22-27061Development and Psychometric Evidence of University Students’ Academic Engagement Scale (USAES) in Mexican College StudentsPLOS ONE

Dear Dr. Parra-Pérez,

Thank you for submitting your manuscript to PLOS ONE. After careful consideration, we feel that it has merit but does not fully meet PLOS ONE’s publication criteria as it currently stands. Therefore, we invite you to submit a revised version of the manuscript that addresses the points raised during the review process. Please submit your revised manuscript by Mar 02 2023 11:59PM. If you will need more time than this to complete your revisions, please reply to this message or contact the journal office at plosone@plos.org. Please include the following items when submitting your revised manuscript:A rebuttal letter that responds to each point raised by the academic editor and reviewer(s). You should upload this letter as a separate file labeled 'Response to Reviewers'.A marked-up copy of your manuscript that highlights changes made to the original version. You should upload this as a separate file labeled 'Revised Manuscript with Track Changes'.An unmarked version of your revised paper without tracked changes. You should upload this as a separate file labeled 'Manuscript'.

We look forward to receiving your revised manuscript.

Kind regards,

Frantisek Sudzina

Academic Editor

PLOS ONE

Journal Requirements:

The research was funded by the Technological Institute of Sonora through its Research

Strengthening Program (Profapi_2022_)

However, funding information should not appear in the Acknowledgments section or other areas of your manuscript. We will only publish funding information present in the Funding Statement section of the online submission form. 

A.A.V-C. Profapi0_2022. Technologic Institute of Sonora (ITSON)

M.U-M Profapi0_2022. Technologic Institute of Sonora (ITSON)

F.I.G-V Profapi0_2022.Technologic Institute of Sonora (ITSON)

URL ITSON:https://itson.mx/Paginas/index.aspx

The funder had no role in study design, data collection and analysis, decusion to publish, or preparation of the manuscript. 

Reviewers' comments:

Reviewer's Responses to Questions

**Comments to the Author**

1. Is the manuscript technically sound, and do the data support the conclusions?

Reviewer #1: Partly

Reviewer #2: Yes

2. Has the statistical analysis been performed appropriately and rigorously? 

Reviewer #1: Yes

Reviewer #2: Yes

3. Have the authors made all data underlying the findings in their manuscript fully available?

Reviewer #1: No

Reviewer #2: No

4. Is the manuscript presented in an intelligible fashion and written in standard English?

Reviewer #1: Yes

Reviewer #2: Yes

5. Review Comments to the Author

Reviewer #1: This study aims to propose and develop a new scale, called University Students’ Academic Engagement Scale (USAES), that is supposed to measure students’ engagement in specific contexts. While the authors have provided some explanations about the procedures of the development of the scale, and the decisions about the data-analytic procedures and results, I have some concerns or suggestions about those choices that are shown below. I also have some suggestions about how to justify the development of the scale instead of focusing on their perceived limitations of the existing scales that measure engagement. In addition, the authors should provide more details regarding the theoretical frameworks, conceptual models, and operational definitions of their measured construct. Please find my comments below.

Line 12: “there are important limitations” � Give some examples and explain what they are.

Line 16: “A sample of 992 Mexican college students…” � Is the purpose of this study focusing on developing the USAES for the Mexican population? Given that this is a newly developed scale, I suggest that the authors should clearly state their target population and how their sample could be regarded as a random, representative sample of the underlying population.

Line 23 “by gender…” � Why is only gender examined to support measurement invariance? How about the influence of other demographic factors or groups to provide a more complete picture about the potential measurement invariance across various groups of participants?

Line 46: “Moreover, regardless of theoretical conceptualizations, current scales tend to capture factors that affect engagement rather than indicators of engagement per se. Such scales comprise broadly worded items (e.g., I do like school) rather than worded to reflect engagement in particular situations.” � I am not an expert in the measurement of engagement, but I believe it is common that psychology researchers conceptualize and operationalize “general” and “situational” psychological constructs. Do you mean that the existing scales focus on measuring “general” engagement, and they do not measure “situational” engagement without providing a context? My quick search seems to show that many studies have examined situational engagement.

Katja Upadyaya, Patricio Cumsille, Beatrice Avalos, Sebastian Araneda, Jari Lavonen & Katariina Salmela-Aro (2021) Patterns of situational engagement and task values in science lessons, The Journal of Educational Research, 114:4, 394-403, DOI: 10.1080/00220671.2021.1955651

Lu, G., Xie, K., & Liu, Q. (2022). What influences student situational engagement in smart classroom: perception of learning environment and students’ motivation. British Journal of Educational Technology. https://doi.org/10.1111/bjet.13204

Line 68: “Second, several scholars [23-26] have explored the test structure using EFA

69 and CFA, finding that NSSE's structure did not hold in their samples.” � Please provide details to support this claim. Do you mean their factor structures, loadings, and/or measurement invariance were not hold across “which” samples?

It is relatively common that the factor structures of a published scale could vary (e.g., 2-factor, 3-factor, second-order factor, etc.) across different samples that are tested by independent researchers due to many factors (e.g., cultural differences). I believe a common practice for examining scale validation is that researchers would test and examine various factor structures that have been found in previous studies and conclude the one that has the best data fit to their study sample, with the goal to accumulate empirical evidence across replications.

Line 69: “it assesses many educational experiences not specifically engagement” � Give examples to support your view.

Line 100: “It seems that given engagement is fundamentally situational; it arises from the interaction of context and individual; scholars must ensure that the scales can capture student engagement within their specific contexts. That is, scales comprising items may not be appropriated, especially if they are interested in exploring how much engagement varies under specific contextual factors. Thus, scale items must be carefully worded to measure engagement in the specific context of higher education”

I believe that this argument is over-stated. First, I am not 100% sure whether the existing literature does not offer any scale that measurement “situational engagement within a context” (I believe there should be some). Second, even if no scale is currently available, previous research may focus on measuring “general” engagement that is conceptualized as a general, psychological trait (e.g., extroversion/introversion of a person based on the Big Five personality). Hence, measuring the construct of “general” engagement should not be regarded as a criticism leading to the current study. Third, I found that there are some other places in which the authors argue that most current, widely employed engagement scales (e.g., UWES-S; lines 87 to 95) are less than optimal because of the mixed results regarding their factor structures. Indeed, validating a scale across different samples in different research settings or labs adheres to the principle of accumulating empirical evidence in science, and it is relatively common among different researchers to observe different factor structures of the same scale (e.g., due to many factors such as culture differences, translation of the scale questions, etc.). In a hypothetical scenario, when future researchers intend to test and validate the proposed USAES based on other samples, it is likely that the USAES would also be found with mixed results (e.g., various factor structures). Or, stated differently, the existing literature does not provide any empirical evidence that the USAES has been comprehensively tested and validated (other than the current sample of n = 992 university students in Mexico) to possess more appropriate psychometric properties (e.g., the same factor structure, high reliability, etc.) than most other current scales (e.g., UWES-S).

In my view, there is no need to criticize “how bad” the current scales are in measuring engagement, and hence, the authors would like to propose and develop a new one. Rather, the tone should be softened, and the authors could directly state that their goal is to develop an engagement scale that measure students’ engagement within specific contexts based on their theoretical framework and conceptual models. Indeed, their paragraphs under the heading: “Theoretical Framework for Measurement Academic Engagement” is a bit misleading because I anticipate that the authors should state their theoretical framework for engagement. For example, what are the existing psychological theories that conceptualize “engagement in a specific context”? How do those theories explain the behavioral, emotional, and cognitive aspects of engagement? Why does the authors select or focus on one theory (if any) that directs them to develop the proposed USAES? After selecting a particular theoretical framework, what conceptual models do they propose (e.g., a conceptual diagram including all the paths that link the antecedents such as academic rigor; line 137) and outcomes?

Regarding “academic rigor” (lines 136 – 139), this section should be expanded. For example, what is the literature review about academic rigor? Why is only one antecedent or predictor selected in this study to predict engagement?

In sum, the current literature review focuses on discussing the authors’ perceived weaknesses of the existing scales that measure engagement without any details (e.g., theoretical frameworks, conceptual models, operational definition, etc.) about their proposed construct. Indeed, I am not 100% sure about the concept or meaning of the proposed construct (should it be “student engagement in specific contexts”? If that is the case, what types of students (e.g., undergraduate, graduate, university students, or others) do they focus on? What “specific contexts” do they refer to?

Line 124 – 125: “Another important issue that has not been addressed in literature is the need to prove that engagement measurement functions similarly across males and females.” �This statement assumes that the variable is gender or a dichotomous view of gender with the categories of males and females only. Line 128: “…found women are equally engaged as their male counterparts…” � The word “women” appears that the authors refer to the “biological sex” of participants. There is some confusion about whether the authors refer to gender, e.g., males/females; gender orientation/tendency that is different from biological sex (e.g., men/women).

Line 146: “members from three public universities located in the north (Sonora)”

How many public universities are there in Mexico? Does the number of 3 ensure that this is a representative sample? As noted above, there is no need to over-state the representativeness, as no study is perfectly or truly based on random sampling. The authors could directly mention their sampling procedure instead of stating that having "a presentative sample from students all over the country".

Line 161: “From these focus group conversations, 25 indicators of academic engagement” � This part misses the details of the qualitative data analysis. E.g., coding, thematic analysis, inter-coder reliability, etc. Please provide the details.

Line 162: “Then, a group of eight experts in higher education” � What are their background, and who are they?

Line 167: “The review by experts ensured high quality and the relevance of the USAES items in alignment with the context and reality of Mexican higher education classrooms”. � This statement seems to be over-stated without the details of the processes that provide scientific evidence and support regarding why and how the experts could ensure “high quality” of the USAES items. “This process led to a considerable reduction of the item pool. In total, 11 of 25 original items of the USAES were removed; only 14 items accomplished a content validity index (CVI) greater than .80 [57]. � Similarly, no evidence has been provided regarding the criteria and processes that delete the 11 items (e.g., based on what approach/method/criteria?)

Line 213: “I do attend all my classes, labs, practices” � Were the items presented in English or Mexican language?

Line 244: “using robust weighted least squares (DWLS)” Why did the authors use DWLS instead of the default estimator maximum likelihood with robust standard error in MPlus? The authors should provide their reasons for choosing a particular estimator.

Bandalos, D. L. (2014). Relative performance of categorical diagonally weighted least squares and robust maximum likelihood estimation. Structural Equation Modeling, 21(1), 102–116. https://doi.org/10.1080/10705511.2014.859510

Li, C. H. (2016). Confirmatory factor analysis with ordinal data: Comparing robust maximum likelihood and diagonally weighted least squares. Behavioral Research Methods, 48, 936–949. https://doi.org/10.3758/s13428-015-0619-7

Line 270: “omega = .70” � Do the authors refer to omega-hierarchical or omega-total? Also, .70 is not high, and it is interpreted as marginal to reasonable reliability. This finding may also suggest that the proposed USAES may not necessarily have better reliability than current scales such as USAES. Or the authors should at least provide the reliability coefficients of other current scales for comparison.

Line 314: “The goodness of fit of second-order factor model (Model B) was not statistically better (ΔSBX2 = 8.54, Δdf = 1, p = .003; ΔBIC = 6.83) than” � It states: “was not statistically better”, but the p value is .003 which is significant at the .05 level (or p < .05).

Reviewer #2: The manuscript is well written. The author/s clearly outlined the justification for the conducted research (i.e., development of the Academic Engagement Scale in Mexican context). Both the theoretical rationale and the presented results make a valuable contribution: provide a new “theoretically-grounded, culturally sensible, and robust” psychometric tool and explain the context of its use in future research as well as educational practice. The study is well designed, and the statistical analyses properly selected and reported. Please find my few minor comments below.

1. The author/s defined three facets of an academic engagement, i.e., behavioral, emotional, and cognitive. However, in the theoretical part, they do not refer to the most common model, known as the ABC model of attitudes (see Ostrom, 1969) that includes these three components. Then it is necessary to specify to what extent academic engagement is derived from the theory of attitudes. This clarification may shed more light on the understanding of this phenomenon.

2. The author/s undertook to study the measurement invariance of the USAES across genders, however, it would be advisable to comment on the equivalence of this tool in the context of its use in different types of schools and in different cultures. Can the developed tool be adapted to other countries in the future? Is it possible to adapt the tool to the conditions of other schools (secondary, primary)? What are the authors' recommendations?

3. Did the authors use the robust Weighted Least Squares (robust WLS) or DWLS (Diagonally Weighted Least Squares) estimation? I am belief that the choice of the DWLS estimator was not the most optimal one but probably robust WLS was used in the study. A brief justification for the choice of the estimator is needed.

4. Since the factor structure showed the second-order engagement factor, the general score should be included in the latent means comparison by gender as well as in the correlations between academic engagement dimensions and academic rigor.

5. The results presented in Table 6 show that affective academic engagement and cognitive academic engagement correlate more strongly with academic rigor than with the behavioral component of academic engagement. Could this have implications for distinguish the three components of the academic engagement.

Ostrom, T. M. (1969). The relationship between the affective, behavioral, and cognitive components of attitude. Journal of Experimental Social Psychology, 5(1), 12–30. https://doi.org/10.1016/0022-1031(69)90003-1

6. PLOS authors have the option to publish the peer review history of their article (what does this mean?). If published, this will include your full peer review and any attached files.

Reviewer #1: No

Reviewer #2: **Yes: **Paweł Jurek

---

## [Author Response · Author response to Decision Letter 0]

29 Mar 2023

Dear Editor

We are thankful to the reviewers for their insightful comments on our manuscript. Their comments and suggestions significantly improve the quality of our work. We have incorporated multiple changes to reflect the suggestions provided by the reviewers. In order to make it easy to identify the changes made within our manuscript, we have highlighted all the additions and changes. You may find a point-by-point response to the reviewers’ concerns in the following paragraphs.

Response to reviewer

Reviewer 1:

Comment

“there are important limitations” Give some examples and explain what they are. See paragraph highlight in yellow (line 16 to 27). 

Comment

“A sample of 992 Mexican college students…” � Is the purpose of this study focusing on developing the USAES for the Mexican population? Given that this is a newly developed scale, I suggest that the authors should clearly state their target population and how their sample could be regarded as a random, representative sample of the underlying population.

Response

We rewrote this section to attend to the reviewer's suggestion. Please see the Participants section. See paragraph highlighted in yellow line 208-214 and line 224. 

Comment

“by gender…” � Why is only gender examined to support measurement invariance? How about the influence of other demographic factors or groups to provide a more complete picture about the potential measurement invariance across various groups of participants?

Response 

Thank you for the insightful suggestion. You are right. Multiple predictors deserve to be further explored in current and emerging studies; however, unlike other demographic factors (race, parental education, socioeconomic status), gender is a demographic factor that consistently reports mixed studies. Such findings came from studies that did not prove the measurement invariance in the scale used. Therefore, ensuring the scale invariance between both genders is critical to bring higher confidence to future research and, hopefully, some consistency in their results. We included in the Limitation section the need for examined invariance across other groups.

Please, see 

• Kinzie J, Thomas AD, Palmer MM, Umbach PD, Kuh GD. Women students at coeducational and women's colleges: How do their experiences compare? J Coll Stud Dev. 2007; 48(2):145–65. https://doi.org/10.1353/csd .2007.0015

• Ní Fhloinn E, Fitzmaurice O, Mac an Bhaird C, O’Sullivan C. Gender differences in the level of engagement with mathematics support in higher education in Ireland. Int J Res Undergrad Math Educ. 2016 Oct; 2(3):297–317. https://doi.org/10.1007/s40753-016-0031-4

• Sontam V, Gabriel G. Student engagement at a large suburban community college: Gender and race differences. Community Coll J Res Pract. 2012 Oct 1; 36(10):808–20. https://doi.org/10.1080/10668926.2010.491998

Please see Limitations section highlighted in yellow line 446 to 450 

Comment

“Moreover, regardless of theoretical conceptualizations, current scales tend to capture factors that affect engagement rather than indicators of engagement per se. Such scales comprise broadly worded items (e.g., I do like school) rather than worded to reflect engagement in particular situation.” � I am not an expert in the measurement of engagement, but I believe it is common that psychology researchers conceptualize and operationalize “general” and “situational” psychological constructs. Do you mean that the existing scales focus on measuring “general” engagement, and they do not measure “situational” engagement without providing a context? My quick search seems to show that many studies have examined situational engagement.

Katja Upadyaya, Patricio Cumsille, Beatrice Avalos, Sebastian Araneda, Jari Lavonen & Katariina Salmela-Aro (2021) Patterns of situational engagement and task values in science lessons, The Journal of Educational Research, 114:4, 394-403, DOI: 10.1080/00220671.2021.1955651

Lu, G., Xie, K., & Liu, Q. (2022). What influences student situational engagement in smart classroom: perception of learning environment and students’ motivation. British Journal of Educational Technology. https://doi.org/10.1111/bjet.13204

Response

You are right. Even though we acknowledge the existence of general and specific scales in our manuscript. This paragraph results at best contradictory. 

We rewrote this paragraph please see paragraph highlighted in yellow line 16 to 27.

Comment

Line 68: “Second, several scholars [23-26] have explored the test structure using EFA

69 and CFA, finding that NSSE's structure did not hold in their samples.” � Please provide details to support this claim. Do you mean their factor structures, loadings, and/or measurement invariance were not hold across “which” samples?

Response

We agree with the reviewer. The factor structures of a published scale could vary. However, confirming the invariance of the scale's internal structure in different samples is still desirable. Samples invariance allows for national and international comparisons with meaningful results.

Please see studies that report different internal structure 

• LaNasa SM.; Olson E., Alleman N. The impact of on-campus student growth on first-year student engagement and success. Res High Educ. 2007:48(8), 941–966 https://doi.or g/10.1007/s11162-007-9056-5 

• LaNasa SM, Cabrera AF, Trangsrud H. The construct validity of student engagement: A confirmatory factor analysis approach. Res in Higher Education. 2009; 50, 315-332. https://doi.org/10.1007/s11162-009-9123-1

• Tendhar C, Culver SM, Burge PL (2013). Validating the National Survey of Student Engagement (NSSE) at a research-intensive university. J Educ Training Stud. 2013; 1(1), 182-193. https://doi.org10.11114/jets.v1i1.70

However, the rewrote the paragraph to attended the reviewer’s suggestion. Please see paragraph highlighted in yellow line 45 to 49.

Comment

Line 69: “it assesses many educational experiences not specifically engagement” � Give examples to support your view.

Response 

We attended the reviewer suggestion (please see paragraph highlighted in yellow line 40 to 43).

Comment

Line 100: “It seems that given engagement is fundamentally situational; it arises from the interaction of context and individual; scholars must ensure that the scales can capture student engagement within their specific contexts. That is, scales comprising items may not be appropriated, especially if they are interested in exploring how much engagement varies under specific contextual factors. Thus, scale items must be carefully worded to measure engagement in the specific context of higher education”

 I believe that this argument is over-stated. First, I am not 100% sure whether the existing literature does not offer any scale that measurement “situational engagement within a context” (I believe there should be some). Second, even if no scale is currently available, previous research may focus on measuring “general” engagement that is conceptualized as a general, psychological trait (e.g., extroversion/introversion of a person based on the Big Five personality). Hence, measuring the construct of “general” engagement should not be regarded as a criticism leading to the current study. Third, I found that there are some other places in which the authors argue that most current, widely employed engagement scales (e.g., UWES-S; lines 87 to 95) are less than optimal because of the mixed results regarding their factor structures. Indeed, validating a scale across different samples in different research settings or labs adheres to the principle of accumulating empirical evidence in science, and it is relatively common among different researchers to observe different factor structures of the same scale (e.g., due to many factors such as culture differences, translation of the scale questions, etc.). In a hypothetical scenario, when future researchers intend to test and validate the proposed USAES based on other samples, it is likely that the USAES would also be found with mixed results (e.g., various factor structures). Or, stated differently, the existing literature does not provide any empirical evidence that the USAES has been comprehensively tested and validated (other than the current sample of n = 992 university students in Mexico) to possess more appropriate psychometric properties (e.g., the same factor structure, high reliability, etc.) than most other current scales (e.g., UWES-S).

In my view, there is no need to criticize “how bad” the current scales are in measuring engagement, and hence, the authors would like to propose and develop a new one. Rather, the tone should be softened, and the authors could directly state that their goal is to develop an engagement scale that measure students’ engagement within specific contexts based on their theoretical framework and conceptual models. Indeed, their paragraphs under the heading: “Theoretical Framework for Measurement Academic Engagement” is a bit misleading because I anticipate that the authors should state their theoretical framework for engagement. For example, what are the existing psychological theories that conceptualize “engagement in a specific context”? How do those theories explain the behavioral, emotional, and cognitive aspects of engagement? Why does the authors select or focus on one theory (if any) that directs them to develop the proposed USAES? After selecting a particular theoretical framework, what conceptual models do they propose (e.g., a conceptual diagram including all the paths that link the antecedents such as academic rigor; line 137) and outcomes?

In sum, the current literature review focuses on discussing the authors’ perceived weaknesses of the existing scales that measure engagement without any details (e.g., theoretical frameworks, conceptual models, operational definition, etc.) about their proposed construct. Indeed, I am not 100% sure about the concept or meaning of the proposed construct (should it be “student engagement in specific contexts”? If that is the case, what types of students (e.g., undergraduate, graduate, university students, or others) do they focus on? What “specific contexts” do they refer to?

Response

We rewrote text to attended the reviewer suggestion. Also, we justified the theoretical approach adopted in the study. Please see the Present Study section paragraph highlighted in yellow line 139 to141 and line 148 to 173. 

Comment

Line 124 – 125: “Another important issue that has not been addressed in literature is the need to prove that engagement measurement functions similarly across males and females.” �This statement assumes that the variable is gender or a dichotomous view of gender with the categories of males and females only. Line 128: “…found women are equally engaged as their male counterparts…” � The word “women” appears that the authors refer to the “biological sex” of participants. There is some confusion about whether the authors refer to gender, e.g., males/females; gender orientation/tendency that is different from biological sex (e.g., men/women).

Response

Thank you for your suggestion. We rewrote this section please see paragraph highlighted in yellow line 112 to 122.

Comment

Line 146: “members from three public universities located in the north (Sonora)”

How many public universities are there in Mexico? Does the number of 3 ensure that this is a representative sample? As noted above, there is no need to over-state the representativeness, as no study is perfectly or truly based on random sampling. The authors could directly mention their sampling procedure instead of stating that having "a presentative sample from students all over the country".

Response 

Thank you for your comment. Actually, what we try to explain in this paragraph is the item-generation process, which was voluntarily supported not only for three public universities located in the north of the country (in Sonora) but also by research participants from universities located in the center (Nayarit) and the south of the country (Chiapas)—12 universities in total. . Please see Participants Section Please see paragraph highlighted in yellow line 208 to 214.

The reviewer must consider that Mexico has more than 100 million inhabitants, with approximately 2000 higher education institutions. These institutions serve different populations from socioeconomic and cultural points of view. We clarify that we use probabilistic sampling that only includes some universities in the country. We include universities from different geographical regions to make the sample more diverse and with greater representativeness. However, we acknowledge that the only thing that guarantees absolute representativeness is to use random sampling, which is highly complicated and expensive in a country like Mexico. We acknowledge this limitation within the limitations section. Please see Limitations Section paragraph highlighted in yellow line 442 to 446.

Comment

Line 161: “From these focus group conversations, 25 indicators of academic engagement” � This part misses the details of the qualitative data analysis. E.g., coding, thematic analysis, inter-coder reliability, etc. Please provide the details.

Respond

We added information to the aforementioned paragraph, please see paragraph highlighted in yellow line 148 to 159 and 160 to 173.

Comment

Line 162: “Then, a group of eight experts in higher education” � What are their background, and who are they?

Response

We rewrote this paragraph please see paragraph highlighted in yellow line 171 to 173

Comment

Line 167: “The review by experts ensured high quality and the relevance of the USAES items in alignment with the context and reality of Mexican higher education classrooms”. � This statement seems to be over-stated without the details of the processes that provide scientific evidence and support regarding why and how the experts could ensure “high quality” of the USAES items. “This process led to a considerable reduction of the item pool. In total, 11 of 25 original items of the USAES were removed; only 14 items accomplished a content validity index (CVI) greater than .80 [57]. � Similarly, no evidence has been provided regarding the criteria and processes that delete the 11 items (e.g., based on what approach/method/criteria?)

Response

We rewrote this paragraph please see paragraph highlighted in yellow line 171 to 173.

We based on ICV values to choose items included in the scale please paragraph highlighted in yellow line 173 to 181.

Comment

Line 213: “I do attend all my classes, labs, practices” � Were the items presented in English or Mexican language?

Response

We added information aimed to clarify the response process within the method section as we thought fits better in this section. See paragraph highlighted in yellow line 253 to 254.

Comment

Line 244: “using robust weighted least squares (DWLS)” Why did the authors use DWLS instead of the default estimator maximum likelihood with robust standard error in MPlus? The authors should provide their reasons for choosing a particular estimator.

Bandalos, D. L. (2014). Relative performance of categorical diagonally weighted least squares and robust maximum likelihood estimation. Structural Equation Modeling, 21(1), 102–116. https://doi.org/10.1080/10705511.2014.859510

Li, C. H. (2016). Confirmatory factor analysis with ordinal data: Comparing robust maximum likelihood and diagonally weighted least squares. Behavioral Research Methods, 48, 936–949. https://doi.org/10.3758/s13428-015-0619-7

Response 

We provided a reason for choosing DWLS estimator. Please see paragraph highlighted in yellow line 263 to 265

 Comment

Line 270: “omega = .70” � Do the authors refer to omega-hierarchical or omega-total? Also, .70 is not high, and it is interpreted as marginal to reasonable reliability. This finding may also suggest that the proposed USAES may not necessarily have better reliability than current scales such as USAES. Or the authors should at least provide the reliability coefficients of other current scales for comparison.

Response

For our data analysis, and based on the literature, we consider that w = .70 indicate an acceptable. However, the values of reliability of us scale reported in the Result section are greater than .80. Please see in the Results section paragraph highlighted in yellow line 335 to 338.

 Comment 

Line 314: “The goodness of fit of second-order factor model (Model B) was not statistically better (ΔSBX2 = 8.54, Δdf = 1, p = .003; ΔBIC = 6.83) than” � It states: “was not statistically better”, but the p value is .003 which is significant at the .05 level (or p < .05).

Response

Based on structural modeling literature, we adopted .001 level to compared model fit. 

Reviewer 2

Comment

From our perspective, academic engagement is a human attitude toward school-related activities, and its measurement should be a tripartite construct comprising affective, behavioral, and cognitive components [17]. According to the ABC model of attitudes [17], individuals’ response consistency toward the three components may be considered as an index of the measured attitude. 

Response

Thank you for your suggestion. We have added information in this regard. Please see paragraph highlighted in yellow line 67 to 73.

Comment

The author/s undertook to study the measurement invariance of the USAES across genders, however, it would be advisable to comment on the equivalence of this tool in the context of its use in different types of schools and in different cultures. Can the developed tool be adapted to other countries in the future? Is it possible to adapt the tool to the conditions of other schools (secondary, primary)? What are the authors' recommendations?

Response

Dear reviewer, thank you for your suggestion that led us toward a deep reflection on this regard. We included in the Conclusion sentence about this suggestion Please see paragraph highlighted in yellow line 452 to 459

Comment

 Did the authors use the robust Weighted Least Squares (robust WLS) or DWLS (Diagonally Weighted Least Squares) estimation? I am belief that the choice of the DWLS estimator was not the most optimal one but probably robust WLS was used in the study. A brief justification for the choice of the estimator is needed.

Response

We included a brief justification of the choice of the DWLS. See in the Data Analysis section line 263 to 265.

Please see 

Bandalos, D. L. (2014). Relative performance of categorical diagonally weighted least squares and robust maximum likelihood estimation. Structural Equation Modeling, 21(1), 102–116. https://doi.org/10.1080/10705511.2014.859510

Bandalos, D. L., & Finney, S. J. (2017). Factor analysis: Exploratory and confirmatory. In G. R. Hancock, L. M Stapleton, & R. O. Mueller (Eds.), The reviewer’s guide to quantitative method in the social science (2nd., pp. 98-122). Routledge.

Finney, S. J., & DiStefano, C. (2013). Nonnormal and categorical data in structural equation modeling. In G. R. Hancock & R. O Mueller (Eds.), Structural equation modeling: A second course (2nd ed., pp. 429-492). Information Age Publishing.

Li, C. H. (2016). Confirmatory factor analysis with ordinal data: Comparing robust maximum likelihood and diagonally weighted least squares. Behavioral Research Methods, 48, 936–949. https://doi.org/10.3758/s13428-015-0619-7

Comment

Since the factor structure showed the second-order engagement factor, the general score should be included in the latent means comparison by gender as well as in the correlations between academic engagement dimensions and academic rigor.

Response

We attended the reviewer suggestion. Please Table 5 and 6.

Comment

The results presented in Table 6 show that affective academic engagement and cognitive academic engagement correlate more strongly with academic rigor than with the behavioral component of academic engagement. Could this have implications for distinguish the three components of the academic engagement.

Ostrom, T. M. (1969). The relationship between the affective, behavioral, and cognitive components of attitude. Journal of Experimental Social Psychology, 5(1), 12–30. https://doi.org/10.1016/0022-1031(69)90003-1

Respond

The reviewer is right, these results attract attention. However, the discriminant validity evidence supports that they are unique constructs. However, it is necessary to review the academic rigor indicators to explore the reasons for these high correlations.

Sincerely

The authors

---

## [Decision Letter · Decision Letter 1]

3 Apr 2023

PONE-D-22-27061R1Development and Psychometric Evidence of University Students’ Academic Engagement Scale (USAES) in Mexican College StudentsPLOS ONE

Dear Dr. Parra-Pérez,

Thank you for submitting your manuscript to PLOS ONE. After careful consideration, we feel that it has merit but does not fully meet PLOS ONE’s publication criteria as it currently stands. Therefore, we invite you to submit a revised version of the manuscript that addresses the points raised during the review process.

We look forward to receiving your revised manuscript.

Kind regards,

Ali B. Mahmoud, Ph.D.

Academic Editor

PLOS ONE

Journal Requirements:

Reviewers' comments:

Reviewer's Responses to Questions

**Comments to the Author**

1. If the authors have adequately addressed your comments raised in a previous round of review and you feel that this manuscript is now acceptable for publication, you may indicate that here to bypass the “Comments to the Author” section, enter your conflict of interest statement in the “Confidential to Editor” section, and submit your "Accept" recommendation.

Reviewer #1: All comments have been addressed

Reviewer #2: All comments have been addressed

2. Is the manuscript technically sound, and do the data support the conclusions?

Reviewer #1: Yes

Reviewer #2: Yes

3. Has the statistical analysis been performed appropriately and rigorously? 

Reviewer #1: Yes

Reviewer #2: Yes

4. Have the authors made all data underlying the findings in their manuscript fully available?

Reviewer #1: No

Reviewer #2: No

5. Is the manuscript presented in an intelligible fashion and written in standard English?

Reviewer #1: Yes

Reviewer #2: Yes

6. Review Comments to the Author

Reviewer #1: I appreciate your flexibility and willingness in incorporating my previous suggestions to your revised study, which has improved a lot (e.g., theoretical approach, sampling and interview procedures, clarity about the choice of the data analysis, etc.)! I have some minor comments below.

Lines 113 – 115: “Although prior research indicates that levels of student engagement may vary significantly by ‘biological sex’ [46-48], some research continues to report mixed results. Whereas a group of scholars [49] found that females are equally engaged as their male counterparts”

Should “biological sex” be changed to “gender”, as you discuss about the differences between males and females?

Lines 335 – 338: � Which omega coefficient did you report (e.g., omega-total, omega hierarchical or others; Trizano-Hermosilla et al., 2021)? Please label your symbol more precisely.

Trizano-Hermosilla, I., Gálvez-Nieto, J. L., Alvarado, J. M., Saiz, J. L., & Salvo-Garrido, S. (2021). Reliability estimation in multidimensional scales: Comparing the bias of six estimators in measures with a bifactor structure. Frontiers in psychology, 12, 508287. https://doi.org/10.3389/fpsyg.2021.508287

Line 310 – 313: “In order to interpret such a correlation, we adopted the guides offered by Cohen [81], which suggest that an r of .10…”

While it is common that researchers use some thresholds (or “t-shirt” sizes) to interpret an effect size, recent studies (e.g., Bakker, et al., 2019) have suggested that this is an inappropriate or inadequate approach (e.g., the criteria for small, medium, and large effect sizes should vary across different disciplines, etc.). I suggest that you should mention about that in the limitation section.

Bakker, A., Cai, J., English, L.D., Kaiser, G., Mesa, V., & Van Dooren, W. (2019). Beyond small, medium, or large: points of consideration when interpreting effect sizes. Educational Studies in Mathematics, 102, 1 - 8.

Line 314: “The goodness of fit of second-order factor model (Model B) was not statistically better (ΔSBX2 = 8.54, Δdf = 1, p = .003; ΔBIC = 6.83) than” � It states: “was not statistically better”, but the p value is .003 which is significant at the .05 level (or p < .05).

Response

Based on structural modeling literature, we adopted .001 level to compared model fit.

Can you provide the citations and discuss the reasons for using .001 as the level of significance for comparing nested models? Using a more stringent .001 (instead of .05) level for testing the differences between two models seem to decrease the likelihood to signal any significant differences of the fit between the two models. In other words, it is more likely to conclude that the two models have the “same” fit (or the two models are identical in terms of their goodness-of-fit levels to your data), unless they have very or super large differences in the population.

Thank you for the opportunity to review this study!

Reviewer #2: I appreciate the Authors’ efforts to improve the manuscript. I think the Authors efficiently responded to my comments. Thus, I find the revised manuscript to be substantially improved and I endorse its publication in the PLOS ONE.

7. PLOS authors have the option to publish the peer review history of their article (what does this mean?). If published, this will include your full peer review and any attached files.

Reviewer #1: No

Reviewer #2: **Yes: **Paweł Jurek

---

## [Author Response · Author response to Decision Letter 1]

2 Jun 2023

Reviewer 1 

Comment

Lines 113 – 115: “Although prior research indicates that levels of student engagement may vary significantly by ‘biological sex’ [46-48], some research continues to report mixed results. Whereas a group of scholars [49] found that females are equally engaged as their male counterparts”

Response

We have addressed the reviewer’s suggestion; please see lines 113-115.

Comment

Lines 335 – 338: � Which omega coefficient did you report (e.g., omega-total, omega hierarchical, or others; Trizano-Hermosilla et al., 2021)? Please label your symbols more precisely here.

Response

We specify that the reported omega is the total omega coefficient (ω).

Comment

While it is common that researchers use some thresholds (or “t-shirt” sizes) to interpret an effect size, recent studies (e.g., Bakker, et al., 2019) have suggested that this is an inappropriate or inadequate approach (e.g., the criteria for small, medium, and large effect sizes should vary across different disciplines, etc.). I suggest that you should mention this in the limitations section.

Response

We acknowledge that there is endless debate on this interpretation. Therefore, we have included this aspect as a limitation in the Discussion section as follows:

Fifth, the reported effect size of the relationships between academic engagement dimensions and academic rigor is based on Cohen´s guides, but it is arbitrary [85]. These results should be compared with those of future studies that use similar designs in Mexican adolescents to test the effect size more accurately.

Comment

Can you provide the citations and discuss the reasons for using .001 as the level of significance for comparing nested models? Using a more stringent .001 (instead of .05) level for testing the differences between two models seem to decrease the likelihood to signal any significant differences of the fit between the two models. In other words, it is more likely to conclude that the two models have the “same” fit (or the two models are identical in terms of their goodness-of-fit levels to your data), unless they have very or super large differences in the population.

Response

We agree with you. Therefore, as suggested by the reviewer, we used a .05. However, this value did not change the interpretation that SBX2 is sensitive to sample size, and we relied on the values of other fit indexes.

---

## [Decision Letter · Decision Letter 2]

19 Jun 2023

Development and Psychometric Evidence of University Students’ Academic Engagement Scale (USAES) in Mexican College Students

PONE-D-22-27061R2

Dear Dr. Parra-Pérez,

We’re pleased to inform you that your manuscript has been judged scientifically suitable for publication and will be formally accepted for publication once it meets all outstanding technical requirements.

Kind regards,

Ali B. Mahmoud, Ph.D.

Academic Editor

PLOS ONE

Additional Editor Comments (optional):

Reviewers' comments:

Reviewer's Responses to Questions

**Comments to the Author**

1. If the authors have adequately addressed your comments raised in a previous round of review and you feel that this manuscript is now acceptable for publication, you may indicate that here to bypass the “Comments to the Author” section, enter your conflict of interest statement in the “Confidential to Editor” section, and submit your "Accept" recommendation.

Reviewer #1: All comments have been addressed

Reviewer #2: All comments have been addressed

2. Is the manuscript technically sound, and do the data support the conclusions?

Reviewer #1: Yes

Reviewer #2: Yes

3. Has the statistical analysis been performed appropriately and rigorously? 

Reviewer #1: Yes

Reviewer #2: Yes

4. Have the authors made all data underlying the findings in their manuscript fully available?

Reviewer #1: No

Reviewer #2: Yes

5. Is the manuscript presented in an intelligible fashion and written in standard English?

Reviewer #1: Yes

Reviewer #2: Yes

6. Review Comments to the Author

Reviewer #1: The authors have appropriately addressed all my previous comments. Thank you and look forward to seeing the published version!

Reviewer #2: I find the revised manuscript to be substantially improved and I endorse its publication in the PLOS ONE.

7. PLOS authors have the option to publish the peer review history of their article (what does this mean?). If published, this will include your full peer review and any attached files.

Reviewer #1: No

Reviewer #2: **Yes: **Paweł Jurek

---

## [Editor Report · Acceptance letter]

4 Aug 2023

PONE-D-22-27061R2 

Development and psychometric evidence of the Academic Engagement Scale (USAES) in Mexican college students 

Dear Dr. Parra-Pérez:

I'm pleased to inform you that your manuscript has been deemed suitable for publication in PLOS ONE. Congratulations! Your manuscript is now with our production department. 

Kind regards, 

on behalf of

Dr. Ali B. Mahmoud 

Academic Editor

PLOS ONE